# Neutrophil-specific transcriptomic profiling reveals a novel signature for active tuberculosis diagnosis

Jie Hu,[1,2] Song Liu,[1,2] Liguo Liu,[1,2] Xingzhu Geng,[1,2] Qianting Yang,[3] Qi Chen,[3] Henan Xin,[1,2] Lei Gao,[1,2] Xiaobing Zhang,[1,2,4] Qi Jin[1,2,4]

**ABSTRACT** Tuberculosis remains a leading cause of mortality worldwide. Rapid and accurate diagnosis of active tuberculosis (ATB) is critical for its effective treatment and disease control. However, current sputum-based detection methods have limited diagnostic coverage for ATB, resulting in missed cases and treatment delay. In this study, we performed neutrophil RNA sequencing (RNA-seq) to identify ATB-specific transcriptional signatures. Through integrated analysis of differentially expressed genes (DEGs) across 173 samples (67 ATB, 42 latent tuberculosis infection [LTBI], and 64 healthy controls [HC]), followed by quantitative Real-time polymerase chain reaction (qPCR) validation using 141 samples (31 ATB, 53 LTBI, 57 HC). A novel 4-gene neutrophil-derived tuberculosis signature (neu-TB) was screened by Lasso model prediction, and further evaluation were performed in two independent cohorts ($n = 332$). The signature achieved exceptional discrimination between ATB and LTBI (area under curve [AUC] = 0.975; 95% confidence interval [CI]: 0.949–1.000; $P < 0.0001$), with 96.8% sensitivity and 92.5% specificity. When distinguishing ATB from HC, the performance remained superior (AUC = 0.980, 91.9% sensitivity, 96.5% specificity). Notably, the signature effectively differentiated ATB from other pulmonary diseases (AUC = 0.969, 95% CI: 0.943–0.995, 93.0% sensitivity, 87.9% specificity). This study represents the first application of immune cell subset-specific transcriptomic profiling to overcome the technical limitations, which are frequently confounded by heterogeneous cellular backgrounds in whole blood or peripheral blood mononuclear cell (PBMC) analyses. The development of this 4-gene neu-TB signature not only establishes a novel methodology for identifying transcriptomic biomarkers but also demonstrates promising clinical applicability for ATB diagnostics.

**IMPORTANCE** The most urgent need in pulmonary TB diagnosis is developing a rapid, non-sputum, biomarker-based diagnostic test. Here, we first identified a novel transcriptomic signature and addressed these challenges through two key innovations: (i) cell-type-specific resolution: by focusing on neutrophils—the most abundant immune cell in TB blood signatures and key mediators of host-pathogen interactions—we reduce biological noise and enhance biomarker precision. (ii) Rigorous multi-cohort validation: the 4-gene neutrophil-derived TB signature (neu-TB) was selected via RNA-seq of purified neutrophils ($n = 141$) and validated by qPCR in two independent cohorts (total $n = 332$), achieving 93% sensitivity/88% specificity in distinguishing active TB from latent infection and non-TB controls.

**KEYWORDS** active tuberculosis, latent infection, transcriptomic signature, neutrophils

Tuberculosis (TB) is an important infectious disease and the leading cause of death from a single infectious agent, worldwide. According to the 2025 World Health Organization (WHO) report, the incidence of TB is 10.7 million worldwide, with an estimated 1.23 million deaths attributed to the diseases (1). TB is caused

**Peer Reviewer** A. Christian Whelen, University of Hawaii, Kaneohe, Hawaii, USA

Address correspondence to Xiaobing Zhang, zhang_xb@ipbcams.ac.cn, Qi Jin, zdsys@vip.sina.com, or Lei Gao, gaolei@ipbcams.ac.cn.

Jie Hu, Song Liu, and Liguo Liu contributed equally to this article. Author order was determined by drawing straws.

The authors declare no conflict of interest.

See the funding table on p. 14.

by *Mycobacterium tuberculosis* and spreads through airborne transmission. Current estimates show that approximately 25% of the global population is infected with *M. tuberculosis*, and the proportion of infections is approximately 20% in China (2, 3). Among those infected, 5%–10% will progress to active TB (ATB) in their lifetime, with the highest risk of disease development occurring within the first 2 years post-infection (4). ATB cases are the primary source of community transmissions. Therefore, rapid detection and identification of ATB are essential for disease prevention and control.

Microbiological confirmation has long served as the gold standard for ATB diagnosis. Despite significant advancements in pathogen detection methodologies, only 64% of the 6.9 million pulmonary TB cases reported worldwide in 2024 were bacteriologically confirmed; the remaining 19%–45% were microbiologically unconfirmed (negative smear, culture, and molecular assays), with the exact proportion varying by country or region (1). This wide variability reflects diverse determinants—background pathogen prevalence, health-care access, availability of molecular diagnostics (e.g., PCR), HIV co-infection, living conditions, and other contextual factors (5–8). Although interferon-gamma release assays (IGRAs) effectively identify *M. tuberculosis* infections, they demonstrate a limited capacity to differentiate between latent tuberculosis infection (LTBI) and ATB. Moreover, the clinical presentation and radiological features of active pulmonary TB frequently overlap with those of other respiratory infections, thereby complicating differential diagnosis. These critical gaps in TB diagnostics underscore the need for novel technological approaches and the identification of validated biomarkers. The latter represents a particularly urgent research priority for precision TB management.

Recent breakthroughs in transcriptomic profiling, most notably RNA sequencing (RNA-seq), have fundamentally transformed our understanding of TB pathogenesis through the systematic delineation of the molecular mechanisms governing host-pathogen dynamics (9, 10). To date, nearly 30 TB transcriptional signatures have been identified through comparative analyses of whole blood and peripheral blood mononuclear cell (PBMC) transcriptomes across ATB, latent LTBI, and healthy control (HC) cohorts (11). Nevertheless, the translational utility of these signatures remains constrained by two critical limitations: population-derived heterogeneity in immune responses and incomplete mechanistic validation of candidate biomarkers (12). The inherent complexity of the peripheral blood substantially complicates the identification of TB-specific transcriptional signatures.

To minimize background interference and refine the screening parameters for TB-specific signatures, we employed two sets of RNA-sequencing data to investigate neutrophil-specific transcriptional profiles of the CD15$^+$ and CD64$^+$ subsets. These two subsets are critical immune mediators in *M. tuberculosis* pathogenesis and were derived from peripheral blood. Previous data reported the ATB-specific profiles of the CD15$^+$ subset (13). The CD64$^+$ neutrophil subset is an established diagnostic marker of bacterial infection, having been consistently shown to expand markedly in patients with systemic infection or sepsis (14). Here, we described the gene expression profiles of the CD64$^+$ subsets among these three groups of isolates. Through systematic comparison of differentially expressed genes (DEGs) in neutrophils across the ATB, LTBI, and HC groups, we identified a novel 4-gene neutrophil-enriched ATB (neu-TB) transcriptional signature. These findings were validated in one cohort and further evaluated in two independent cohorts. The signature demonstrated high sensitivity and specificity in distinguishing individuals with ATB from those with LTBI, healthy controls (HC), and other pulmonary diseases such as pneumonia and lung cancer.

## MATERIALS AND METHODS

### Study participants and sample collection

A total of 646 clinical samples were analyzed in this study, which were from 271 ATB cases, 150 LTBI cases, 33 non-tuberculous mycobacterial infection (NTB) cases, and 192

HC. Samples were prospectively collected from five clinical centers located across four Chinese cities (Beijing, Shenzhen, Zhengzhou, and Suzhou) between 2016 and 2024.

ATB cases met the following criteria: (i) bacteriologically confirmed diagnosis through sputum smear positivity, mycobacterial culture growth, or molecular test positivity, (ii) absence of co-existing infections, and (iii) ≤15 days of anti-TB treatment initiation. LTBI was defined as a positive IGRA result with normal chest radiographic findings. Bacterial pneumonia (BPN) and lung cancer were classified into the NTB disease group. BPN was characterized by (i) clinical manifestations of respiratory infection (fever, productive cough, pleuritic pain), (ii) radiographic infiltrates on chest imaging, (iii) laboratory evidence of leukocytosis with neutrophilic predominance, and (iv) negative IGRA results. Individuals with lung cancer were identified on a clinical and diagnostic basis. HCs demonstrated a negative IGRA status, normal chest imaging, and no clinical evidence of active infections. The study protocol was approved by the Ethics Committee of the National Institute of Pathogen Biology of Chinese Academy of Medical Sciences.

## Transcriptomic profiling of equal numbers of CD64$^+$ cells from clinical samples

In this study, equal numbers of CD64$^+$ cells were isolated from each of 114 participants (Discovery cohort D1: 52 ATB, 19 LTBI, 43 HC) and subjected to RNA sequencing. Two milliliters of fresh peripheral blood was collected in EDTA-coated vacutainer tubes (BD Biosciences, USA). CD64$^+$ neutrophils were enriched using fluorescence-activated cell sorting (Fig. S1). After erythrocyte lysis, whole blood samples were stained with the following fluorescent antibodies: anti-CD45 PerCP-Cy5.5 (clone HI30; BD Biosciences), anti-CD3 PE-Cy7 (clone SK7; BD Biosciences), and anti-CD64 FITC (clone 10.1; BD Biosciences). To minimize neutrophil RNA degradation, we adopted a single-tube "all-in-one" strategy: A total of 150 CD45$^+$CD3$^-$CD64$^{dim}$ cells per sample were directly sorted into RNase-Inhibitor-containing water and immediately processed for cDNA library construction. cDNA libraries were constructed using the SMARTer Stranded Total RNA-Seq Kit v3—Pico Input Mammalian (Takara Bio, USA) according to the manufacturer's specifications. High-throughput sequencing was conducted on an Illumina NovaSeq 6000 system (Illumina, USA) using 150 bp paired-end chemistry.

## Data processing and differential expression analysis

Differential gene expression was analyzed separately for two RNA-seq data sets: the newly generated CD64$^+$ subset profiles and our previously published CD15$^+$ subset data (Discovery cohort D2: 15 ATB, 23 LTBI, 21 HC) (13). RNA-seq libraries of CD15$^+$ subset were generated with the SMARTer Stranded Total RNA-Seq Kit v2 (Takara) from 0.5 mL of participant peripheral blood. Sequencing was performed on the NovaSeq 6000 platform. Raw sequencing data were subjected to quality filtering using Trimmomatic software (15). This included removal of adapter sequences (introduced during library preparation), low-quality reads (resulting from sequencing errors), and short reads (<30 bp). Cufflinks software was used to calculate the Fragments Per Kilobase of exon model per million mapped fragments (FPKM) for each gene (16). Samples containing five or more genes with FPKM values >100 were selected for downstream analysis. Pairwise comparisons of gene expression were conducted using DESeq2 to compute fold change (FC), $P$-values, and false discovery rate (FDR). Differentially expressed genes (DEGs) in the neutrophil transcriptome were identified using the following criteria: FPKM >100, $P < 0.05$, |log2FC| > 0.5 or < −0.5. The functional analysis of DEGs was performed using STRING v11, and the resulting protein-protein interaction (PPI) network was visualized with Cytoscape (17, 18).

## CD15$^+$ neutrophil enrichment and RNA processing

Peripheral-blood CD15$^+$ neutrophils were enriched with Dynabeads CD15 (Thermo Fisher Scientific) from participants in the verification cohort (31 ATB, 53 LTBI, 57 HC), evaluation cohort 1 (87 ATB, 28 LTBI, 30 HC), and evaluation cohort 2 (86 ATB, 27 LTBI, 41 HC, 33

NTB). Briefly, 1 mL of fresh peripheral blood was collected in EDTA-coated tubes, diluted with two volumes of phosphate-buffered saline, and incubated with 25 µL of pre-washed Dynabeads CD15 (Thermo Fisher) for 20 min at 8°C. $CD15^+$ neutrophil-bead complexes were isolated using a magnet, followed by three washes with PBS to remove unbound cells. This protocol consistently yields preparations in which >90% of the enriched cells are $CD15^+$ neutrophils from whole-blood samples (Fig. S2). Total RNA was purified from enriched neutrophils using the RNeasy Plus Mini Kit (Qiagen, Hilden, Germany), which includes a gDNA eliminator column to remove genomic DNA contamination. Complementary DNA (cDNA) synthesis was performed using SuperScript VILO MasterMix (Thermo Fisher Scientific) according to the manufacturer's instructions.

## The performance of qPCR

qPCR validation of host target-gene expression was carried out with TaqMan Fast Advanced Master Mix (Thermo Fisher Scientific) on the Applied Biosystems QuantStudio 7 Pro Real-Time PCR System (Thermo Fisher Scientific). cDNA templates were amplified using gene-specific or reference gene primers (Table S1), and threshold cycle (Ct) values were quantified for both target and reference genes. The relative quantification (RQ) of gene expression was calculated using the $2^{-\Delta\Delta Ct}$ method, representing fold changes in the experimental group relative to the control group. The TaqMan assay probes were used for target gene detection. Candidate reference genes were selected based on a transcriptome-wide analysis of all enrolled samples. MYO1F demonstrated superior stability compared to ACTB and GAPDH in neutrophils and was subsequently used as the reference gene for this study.

## Calculating score of TB diagnostic signatures

A logistic regression model was applied to neutrophil gene expression data to calculate the combined score of the 4-gene neu-TB signature for each isolate. The probability of ATB diagnosis ($P_{TB}$) was calculated by logistic function as follows:

$$P_{TB} = \frac{1}{1 + e^{(\beta_0 + \beta_1 X_1 + \beta_2 X_2 + \beta_3 X_3 + \beta_4 X_4)}}$$

where β0, β1, β2, β3, and β4 represent the model coefficients, and $X1$, $X2$, $X3$, and $X4$ correspond to the expression levels of the target genes. A predefined PTB threshold was used to classify samples as positive or negative for ATB.

## Statistical analysis

All statistical analyses were conducted using R statistical environment (https://www.r-project.org/, version 4.3.3) and IBM SPSS Statistics (https://www.ibm.com/products/spss-statistics, version 29.0.1.0). Two machine learning algorithms, LASSO regression and random forest, were employed for feature selection (19, 20). All analyses were implemented in Python (version 3.13.7; https://www.python.org). The diagnostic performance was evaluated by calculating the area under the receiver operating characteristic curve (AUC), with the corresponding sensitivity and specificity determined using Youden's index. Intergroup differences in AUC values were assessed using the Wilcoxon rank-sum test (Mann-Whitney $U$ test).

## RESULTS

### Study design for screening neu-TB signature

The study design comprised two discovery cohorts, one validation cohort, and two independent evaluation cohorts (Fig. 1). In total, clinical isolates from 646 participants belonged to five distinct cohorts. Epidemiological characteristics and bacterial detection data are summarized in Table 1. Transcriptomic profiling of two neutrophil

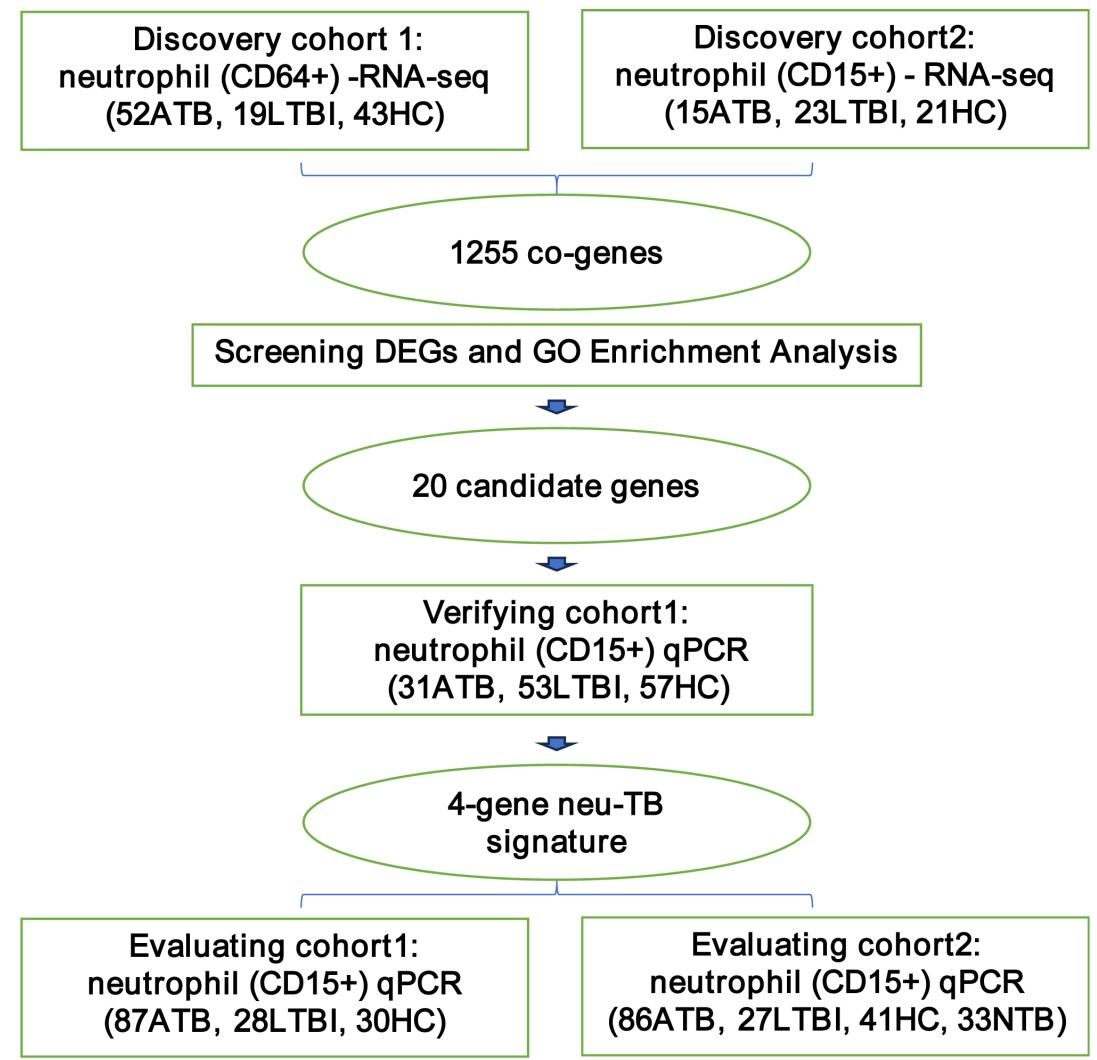

**FIG 1** Experimental workflow of the study. The study design comprised five independent cohorts. RNA sequencing of two neutrophil subtypes revealed 1,255 overlapping genes (co-genes). Differential expression analysis (DEGs) and Gene Ontology (GO) enrichment analysis identified 20 candidate genes. A 4-gene signature derived from qPCR validation was further assessed in two external cohorts.

subpopulations (CD64$^+$ and CD15$^+$ neutrophils) was conducted in the discovery cohorts. Twenty candidate target genes were identified from a shared DEG set between the two cohorts. In the validation cohort, the RNA-seq results for these candidates were confirmed by quantitative real-time PCR (qPCR), and a LASSO regression model was applied to refine the selection, yielding an optimized 4-gene combination of MYL12A, EIF1, GBP5, and SRSF5. The diagnostic performance of this 4-gene neu-TB signature was evaluated in two independent cohorts.

## The characterization of differentially expressed genes in CD64$^+$ neutrophils from ATB patients

RNA-seq analysis provided data on a total of 3,485 messenger RNAs (mRNAs) in the CD64$^+$ subset (cohort D1, discovery cohort 1). A total of 853 genes ($P < 0.05$) exhibited statistically significant differential expression, of which 593 DEGs (log2FC < −0.5 or > 0.5) were obtained from the comparison among ATB, LTBI, and HC groups. Compared with HC, there were 423 DEGs exhibited in ATB, of which 223 were upregulated and 200 were downregulated. Compared with LTBI, a total of 245 DEGs (142 upregulated and 103 downregulated) were identified in ATB. In contrast, LTBI showed 94 DEGs relative to HC

**TABLE 1** Overview of the included cohorts in this study

| Parameter | Discovery | | Verification: | Evaluation | |
|---|---|---|---|---|---|
| | D1 cohort | D2 cohort | V cohort | E1 cohort | E2 cohort |
| Number | 114 | 59 | 141 | 145 | 187 |
| Gender | | | | | |
| Male (%) | 54 (47.4) | 31 (52.5) | 85 (60.3) | 108 (74.5) | 107 (57.2) |
| Female (%) | 60 (52.6) | 28 (47.5) | 56 (39.7) | 37 (25.5) | 80 (42.8) |
| Age | | | | | |
| 18–60 (%) | 110 (96.5) | 59 (100) | 128 (90.8) | 81 (55.9) | 108 (57.8) |
| <18 or >60 (%) | 4 (3.5) | 0 | 13 (9.2) | 64 (44.1) | 79 (42.2) |
| Group | | | | | |
| ATB[a] | 52 | 15 | 31 | 87 | 86 |
| Smear positive (%) | 44 (84.6) | 9 (60) | 11 (35.8) | 34 (39.1) | 33 (38.4) |
| Culture positive (%) | 46 (88.5) | 6 (40) | 25 (80.6) | ns[c] | ns |
| Nuclear positive (%) | 36 (69.2) | 14 (93.3) | 24 (77.4) | 52 (59.8) | 57 (66.3) |
| LTBI | 19 | 23 | 53 | 28 | 27 |
| HC | 43 | 21 | 57 | 30 | 41 |
| NTB[b] | | | | | 33 |
| PN | | | | | 9 |
| CA | | | | | 20 |
| Fungi/pleural effusion | | | | | 4 |

[a]The detection rates of the three sputum-based methods were analyzed in parallel.
[b]The NTB samples consist of PN, CA, and fungi/pleural effusion.
[c]ns, no sample.

(55 upregulated and 39 downregulated). Heatmap analysis further demonstrated that the gene-expression profile of the ATB group was clearly distinct from those of the LTBI and HC groups (Fig. 2).

Among 593 DEGs, 295 (49.7%) were upregulated in ATB samples. Among these, 189 DEGs (64.1%) were unique to the ATB vs HC or ATB vs LTBI comparison, 70 (23.7%) were shared between both contrasts, 23 (7.8%) were upregulated in both comparisons, and 13 (4.4%) were upregulated in ATB vs LTBI but downregulated in LTBI vs HC. Among the 263 downregulated DEGs in ATB, 206 (78.3%) were identified in either ATB vs HC or ATB vs LTBI, 45 (17.1%) were downregulated in both contrasts, 10 (3.8%) were downregulated in ATB vs LTBI but upregulated in LTBI vs HC, and 2 (0.8%) were downregulated in both ATB vs LTBI and LTBI vs HC. Overall, the difference in transcriptomic profiles among ATB, LTBI, and HC revealed a pattern of cumulative changes in specific genes. This indicates that the ATB transcriptome underwent significant reprogramming, predominantly attributed to the upregulation of functionally critical genes.

## Screening candidate signatures from DEGs

To investigate the relevance of transcriptome profiles between these two neutrophil subtypes, we analyzed 1,255 expressed genes detected across both data sets (referred to as co-genes), representing 36.01% of the total gene count in the CD64$^+$ subset and 52.25% in the CD15$^+$ subset. Among these co-expressed genes, differential expression analysis revealed 298 DEGs (23.7%) in CD64$^+$ neutrophils in the ATB vs HC comparisons and 152 DEGs (12.1%) in the ATB vs LTBI comparisons. We identified 383 DEGs (30.5%) in CD15$^+$ neutrophils for ATB vs HC and 377 DEGs (30.0%) for ATB vs LTBI. Functional enrichment analysis revealed a substantial overlap in biological processes between the two neutrophil subtypes. The top 30 terms of functional annotations were predominantly associated with innate immune mechanisms, including immune regulation, activation of signal transduction, cytokine biosynthesis, and cell surface receptor signaling pathways (Fig. 3). Based on this finding, we selected 20 DEGs from these DEGs as critical genes for qPCR validation (Table S2). The selection criteria were as follows: differential expression magnitude (log2FC > 0.5), intergroup statistical significance (ATB

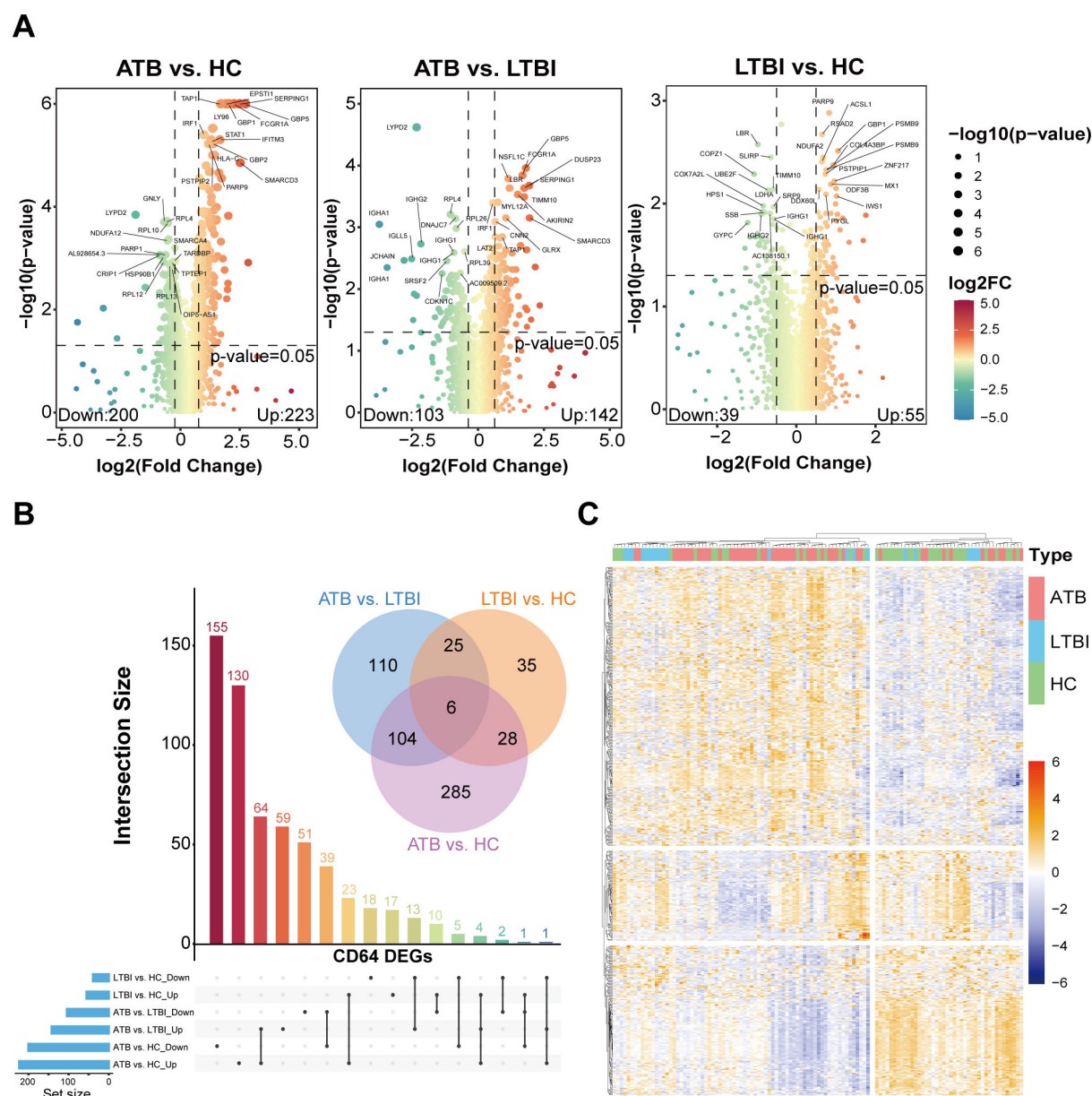

**FIG 2** Analysis of RNA-seq in CD64$^+$ neutrophils from ATB, LTBI, and HC. (A) Volcano plots of DEGs for three kinds of comparison: x axis, log2 fold change of ATB compared with HC; y axis, −log10 of P value; red dots represent upregulated DEGs, and blue dots represent downregulated DEGs. Marked DEGs refer to the most significant DEGs for comparisons between ATB vs HC, ATB vs LTBI, and LTBI vs HC. (B) The number of DEGs covers comparisons among these three groups of samples. (C) Hierarchical clustering heatmap of DEGs in all samples. Rows represent genes, and columns represent samples. Relative levels of gene expression are displayed by a color scale, with red or yellow indicating high expression and blue indicating low expression.

vs HC or ATB vs LTBI, $P < 0.001$), and functional relevance identified through integrated analysis of Gene Ontology (GO) terms and KEGG pathways (21). Functional prioritization specifically emphasized immune response pathways, ribosomal protein biosynthesis, and macromolecular complex assembly mechanisms (Fig. 3).

## Validation of candidate gene expression and screening of transcriptomic signatures

In the validation cohort, qPCR was used to detect the expression of 20 DEGs across the three groups (ATB, LTBI, and HC). The statistical characteristics of these candidate genes were categorized into four distinct groups according to their comparative significance

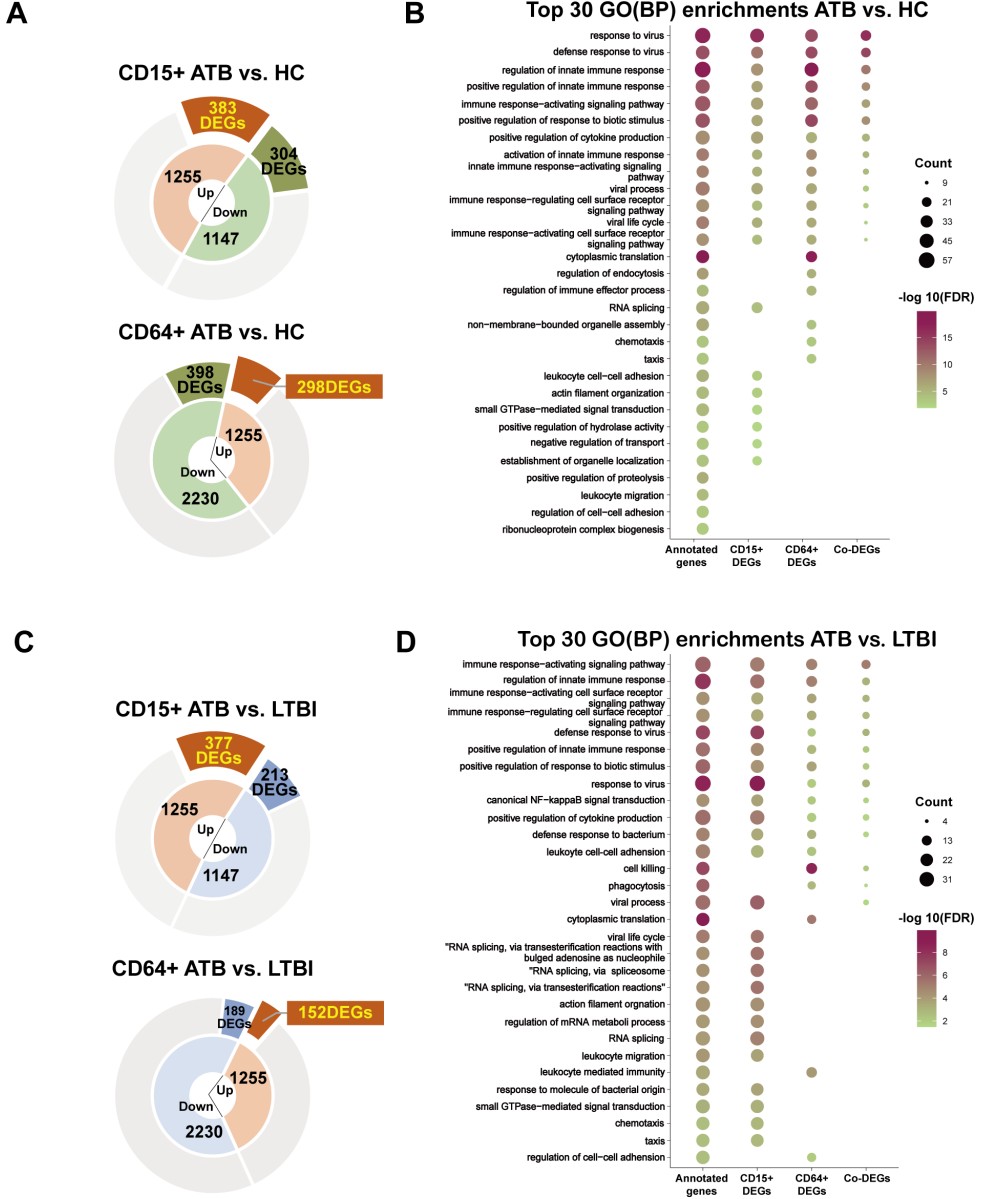

**FIG 3** Gene annotation statistics and enriched functional profiles of co-expressed DEGs in CD15[+] and CD64[+] neutrophil subtypes. (A and C) The counts of DEGs identified from the ATB vs HC and ATB vs LTBI comparisons in each subset are indicated in yellow. (B and D) The top 30 significantly enriched gene ontology (GO) terms for DEGs in CD15[+] and CD64[+] neutrophils are displayed for the ATB vs HC (upper panels) and ATB vs LTBI (lower panels) comparisons, ranked by false discovery rate (FDR).

profiles. Ten genes demonstrated concurrent differential expression in both the CD64[+] and CD15[+] neutrophil subsets. Exclusive differential expression patterns were observed for five genes in the CD64[+] subset and three genes in the CD15[+] subset. In addition, two genes (NFE2 and THEMIS2) were designated as negative controls for experimental validation.

Gene expression levels were determined using qPCR. The result displayed that there were 16 genes exhibited statistically significant differences ($P< 0.05$) as comparing ATB cases to both the HC and LTBI groups (Fig. 4A). Notably, all these DEGs showed upregulated expression patterns in ATB samples compared to the other groups. Furthermore, two negative control genes, two CD64[+] subset-specific DEGs, and one overlapping

DEG shared between subsets exhibited no statistically significant differential expression across the groups. These findings validated the reliability of the DEG analysis.

Next, the random forest model was used to evaluate feature importance in a simulated classification task involving these 20 genes. The top five predictive features were GBP5, MYL12A, STAT1, EIF1, and SRSF5. Subsequent LASSO regression analysis prioritized MYL12A, EIF1, GBP5, and SRSF5 for further analysis. Combinations of these genes effectively discriminated between ATB and HC, ATB and LTBI, and LTBI and HC (Fig. 4B). Notably, individual genes (MYL12A, EIF1, GBP5, and SRSF5) exhibited AUCs ranging from 0.77 to 0.96. The highest diagnostic performance was obtained when the expression of these four genes was combined, achieving an AUC of 0.984 [0.969–1.000] compared to the expression of a single gene (Fig. 4C).

## Evaluating the discriminatory ability of the 4-gene neu-TB signature

We employed receiver operating characteristic (ROC) curve analysis to assess the discriminatory performance of the 4-gene neu-TB signature in one validation cohort (V) and two independent evaluation cohorts (E1 and E2). For the validation and E1 cohorts, AUC values with corresponding sensitivity and specificity were calculated using binary logistic regression analysis in SPSS under three comparative conditions: ATB vs HC, ATB vs LTBI, and ATB vs combined HC+LTBI groups. In the cohort E2 analysis, additional parameters were computed for ATB vs NTB, including BPN and cancer (CA), along with the three aforementioned comparison models, as shown in Fig. 5.

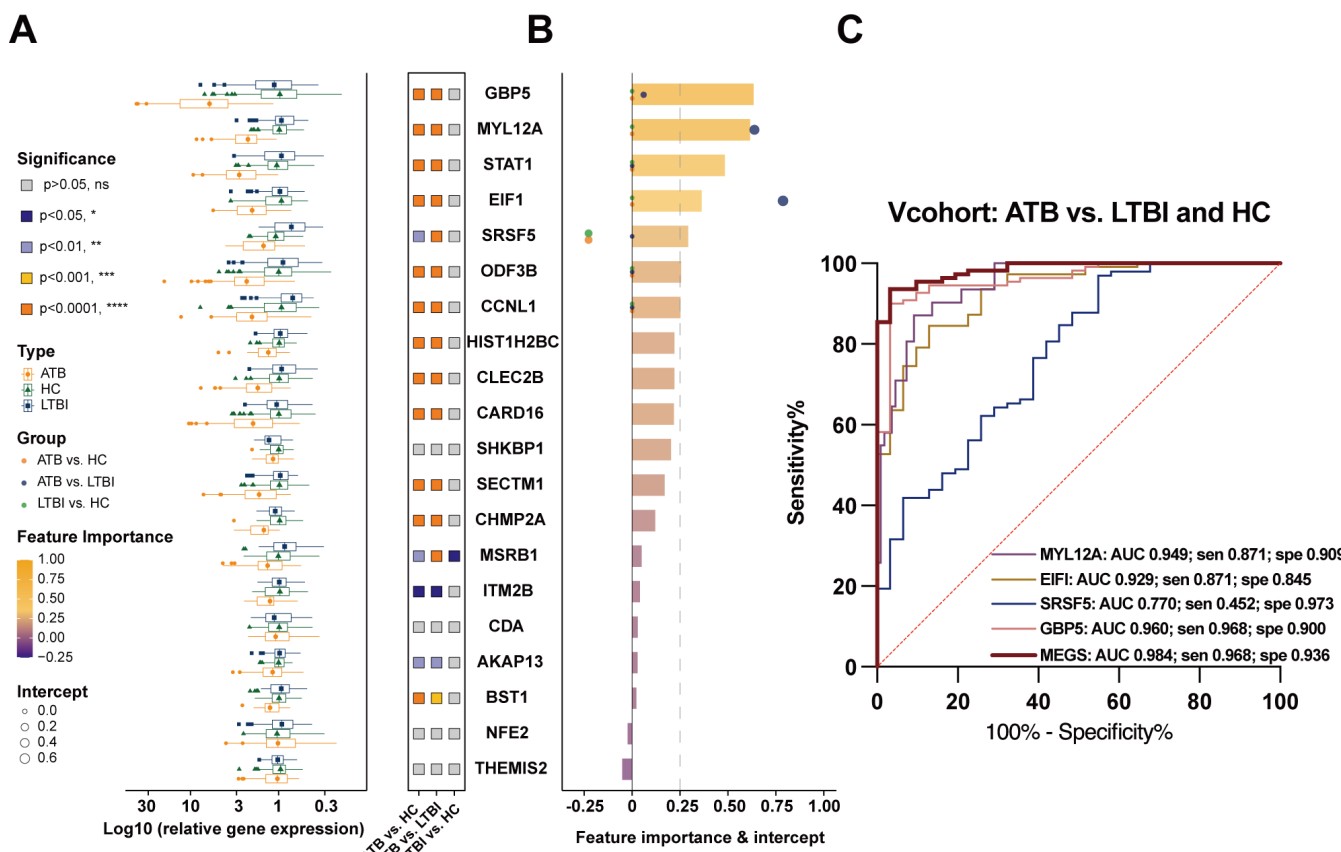

**FIG 4** Statistical analysis of qPCR data and feature-importance ranking for the 20 selected genes. (A) Statistical comparison of qPCR results for the 20 genes across groups, analyzed using the Kruskal-Wallis test. (B) Analysis of feature importance and model intercept for the 20 genes, highlighting GBP5, MYL12A, STAT1, EIF1, and SRSF5 as the most important features. Among these genes, GBP5, MYL12A, EIF1, and SRSF5 were selected via LASSO regression modeling. (C) The combined expression levels of these genes demonstrated the highest discriminative ability between ATB and other groups, as assessed by the model. This figure provides a comprehensive overview of gene expression patterns and their relevance in distinguishing ATB from control groups.

For ATB vs HC, the neu-TB signature exhibited remarkably high AUC values, sensitivities, and specificities across all three cohorts. The AUC values were 0.995 (95% confidence interval [CI]: 0.986–1.000, $P < 0.0001$), 0.986 (95% CI: 0.976–1.000, $P < 0.0001$), and 0.978, respectively. The sensitivities ranged from 0.908 (cohort E1) to 0.968, while the specificities ranged from 0.965 to 1.000. For discrimination between ATB and LTBI, the diagnostic performance remained robust, with AUC values ranging from 0.958 (95% CI: 0.925–0.994) (cohort E1) to 0.975 (95% CI: 0.948–1.000) (cohort E2). The sensitivities varied between 0.885 (cohort E1) and 0.968 (cohort V), and the specificities consistently exceeded 0.925 for all three cohorts. For the discrimination of ATB vs HC and LTBI, the AUC values were 0.973–0.984, with sensitivities ranging from 0.897 to 0.968 and specificities between 0.926 and 0.948. For the discrimination of ATB from NTB, the AUC value was 0.967 (95% CI: 0.943–0.995), accompanied by a sensitivity of 0.930 and a specificity of 0.897.

The performance of the pTB prediction model was evaluated using a predefined cut-off value for positive predictions, and the predicted results were compared with the actual diagnostic status of individuals. As shown in Fig. 4B, the consistency between the predicted and actual classifications exceeded 90% for the ATB group. Among the tested isolates, the positive prediction rate was higher for smear-positive isolates, reaching 100% in cohorts V and E2. For nucleic acid-positive tuberculosis (nuclear positive TB), the positive prediction rate ranged from 88.5% to 96.5%.

In contrast, the positivity rates for the HC and LTBI groups remained below 10%, with the positivity rate in the LTBI group being approximately double that of the HC group. For NTB isolates, the false-positive rate was 15.2% among the 33 NTB patients, which included three isolates from patients with cancer and two isolates associated with other diseases. These results highlight the robustness of the pTB prediction model in distinguishing active TB from other conditions while also demonstrating its specificity in minimizing false positives across diverse clinical scenarios.

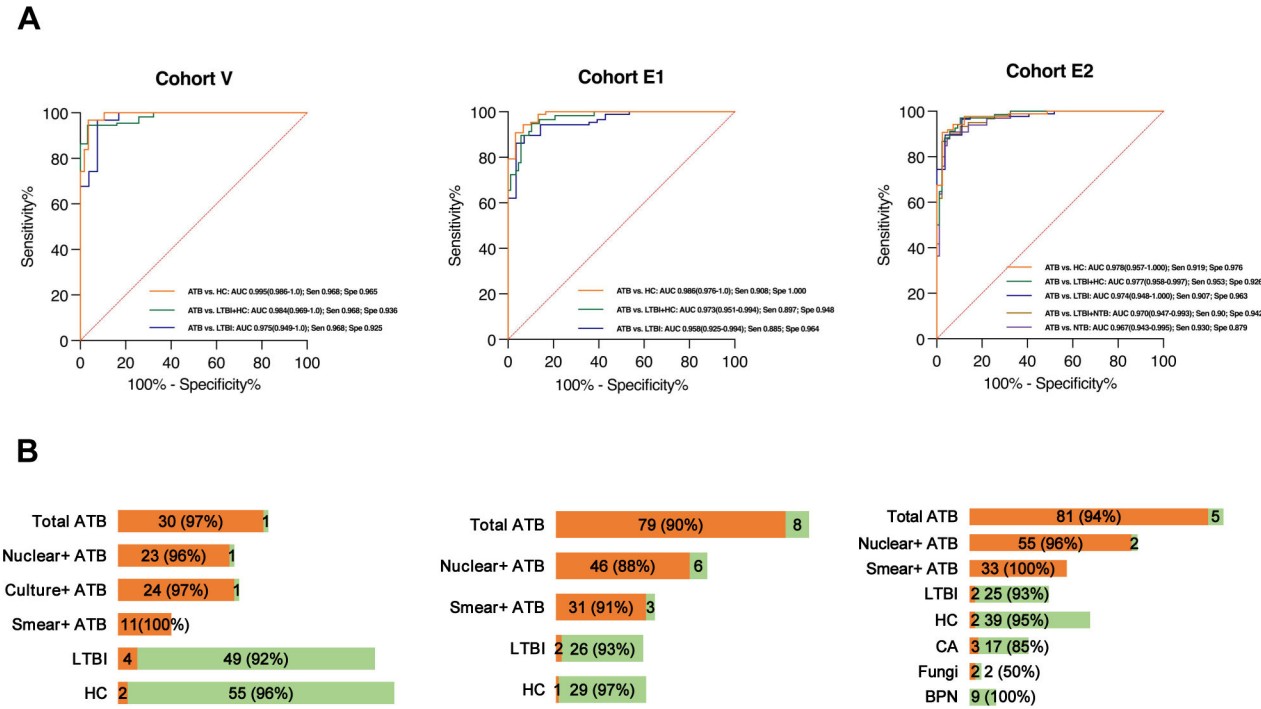

**FIG 5** Evaluation of the 4-gene neu-TB signature as a diagnostic marker for active tuberculosis (ATB). (A) Receiver operating characteristic (ROC) curve analysis and corresponding areas under the curve (AUC) with 95% confidence intervals (CI) demonstrating this signature's ability to distinguish active TB from other groups (HC, LTBI, HC+LTBI, NTB). (B) Positive predictive values across sample types, highlighting the neu-TB signature's specificity and performance in differentiating ATB from other conditions.

## DISCUSSION

The advancement of novel TB diagnostic technologies represents a critical strategy for enhancing diagnostic efficiency and improving disease prevention and control. Sputum-based diagnostics lack sensitivity to detect TB in patients who are smear- or culture-negative yet IGRA- or TST-positive, nor in those unable to expectorate. Therefore, WHO actively endorses development efforts to discover non-sputum-based biomarkers, such as host response signatures. Meanwhile, clear technical standards and evaluation frameworks have been established for distinct product categories, such as triage testing, definitive diagnosis, disease progression prediction, and treatment response monitoring (22). According to this report, triage products must demonstrate minimum sensitivity and specificity thresholds of 90% and 70%, respectively. For TB diagnostic products, the required sensitivity should be ≥65% overall. Meanwhile, this parameter must exceed 98% in patients with smear-positive culture-confirmed pulmonary TB. In addition, the specificity must meet or exceed 98% across all the tested populations.

Recent advances can constitute next-generation diagnostic candidates for improving the diagnostic efficiency in HIV-infected patients and children (23, 24). The response of the peripheral circulatory system has emerged as a central research focus in TB pathogenesis. This is due to its capacity to integrate molecular signatures that elucidate systemic immune sensing, regulatory networks, and downstream effector mechanisms (25, 26). By leveraging blood transcriptomics, researchers can systematically describe infection-driven perturbations in immune gene expression networks, thereby facilitating identification of non-sputum-based molecular signatures (27–30). However, substantial heterogeneity persists among reported blood transcriptomic signatures across studies. For instance, cumulative studies have identified 109 DEGs across eight signatures; however, 15 DEGs (13.8%) emerged in two or three signatures simultaneously (31). Meanwhile, the accuracy of reported host response signatures needs further improvement to meet WHO requirements for sensitivity and specificity (32). Thus, novel strategies are warranted to refine these signatures.

In this study, we applied a new strategy to minimize the interference of background factors. By focusing on the transcriptomic profile of neutrophils, a TB-specific signature was identified and evaluated in three independent cohorts. As neutrophils play pivotal roles in both infection containment and disease pathogenesis, emerging evidence has highlighted specific neutrophil subsets in TB immunopathology (33, 34). For neutrophils, the subsets of CD15$^+$ and CD64$^+$ have been frequently reported in multiple studies about TB diagnosis. Notably, the expansion of low-density neutrophils, a distinct subpopulation expressing surface CD15, has been observed to be significantly elevated in ATB patients (35). CD64 (FcγRI) demonstrates functional pluripotency, mediating phagocytosis, respiratory burst activity, and antibody-dependent cellular cytotoxicity across monocytic, macrophagic, and granulocytic lineages. This solidifies its status as a bacterial infection biomarker (36). Comparative analyses have revealed significant increases in both absolute counts and proportional representation of CD64$^+$ neutrophils in active TB cases vs healthy controls (37, 38). Flow-cytometric quantification in cohort D1 showed that the CD64$^+$ compartment was significantly expanded in both ATB and LTBI isolates compared with HC samples (Fig. S3). Further analysis revealed robust co-expression of CD15 and CD64 on neutrophils in ATB isolates. Dual staining of peripheral-blood leukocytes demonstrated that >90% of CD64$^+$ cells co-expressed CD15, and conversely, the majority of CD15$^+$ cells fell within the CD64$^+$ gate (Fig. S4).

Our transcriptomic TB signatures were screened by employing RNA-seq data from these two subsets of neutrophils using two different approaches. First, to ensure the detection of absolute differences in gene expression, we performed comparative transcriptome analysis among three groups of samples using flow cytometry to sort cell populations to ensure an equal number of CD64$^+$ subset cells (a minimum of 150 cells with CD45$^+$CD3$^-$CD64$^+$). Neutrophils are notoriously labile in the peripheral circulation, enriching only a small number of cells limited degradation and ensured experimental precision. Meanwhile, we conducted a volume-adjusted transcriptional analysis of

peripheral blood CD15$^+$ neutrophils, which are the dominant circulatory neutrophil subset using an equal volume of whole blood for each sample (13). KEGG analysis of the DEGs revealed a shared transcriptional signature between the two subsets: 60% (ATB vs HC) and 69% (ATB vs LTBI) of CD64$^+$ DEGs were concordantly identified in CD15$^+$ cells (Fig. S5), indicating that both subset-specific analyses converge on the same analytical target. By combining transcriptomic data sets of these two subsets, we aimed to identify ATB-specific signatures. These signatures accurately characterize neutrophil functional states in the systemic circulation.

This neu-TB signature is implicated in critical physiological processes, including innate immune regulation, inflammatory responses, and initiation of protein translation. MYL12A encodes myosin light chain 12A, the phosphorylation of which regulates smooth muscle and non-muscle cell contraction. Previous studies have indicated that MYL12A phosphorylation induces actin rearrangement and enhances macrophage migration and invasion during *Salmonella* infections (39). Similarly, GBP5, a guanylate-binding protein, activates NLRP3 inflammasome assembly and has been linked to innate immunity and inflammatory pathways (40). Notably, GBP5 has been identified as a transcriptional signature in several TB-related studies (31, 41). EIF1, which encodes eukaryotic translation initiation factor 1, modulates translation initiation through its RNA binding activity. Recent evidence highlights its dual localization in the cytoplasm and nucleus, with nuclear eIF1 release reportedly influencing start-codon selectivity during mitosis, thereby regulating the balance between mitotic slippage and apoptosis (42). SRSF5, a serine/arginine-rich splicing factor, is involved in mRNA splicing, nuclear export, and translation. Dysregulation of splicing factors such as SRSF5 can induce aberrant RNA processing and promote tumorigenic phenotypes including proliferation, migration, and apoptosis resistance. Emerging evidence suggests that SRSF5 is a potential diagnostic biomarker for small cell lung cancer and pleural metastatic malignancies (43). Thus, the four-gene combination demonstrated superior accuracy in distinguishing ATB from other groups.

Using this neu-TB signature, we could discriminate ATB cases from samples of LTBI and HC samples. Notably, the positive predictive value reached 100% in smear-positive TB samples across one validation cohort (V) and one evaluation cohort (E2), as shown in Fig. 4B. ROC curve analysis revealed high diagnostic accuracy, with sensitivities ranging from 88.5% to 96.8% and specificities between 87.9% and 100%. These performance metrics satisfied the WHO minimum requirements for non-sputum-based triage tests (sensitivity >65% and specificity ≥90%).

NTB patients constitute confounding cases in TB diagnosis. The neu-TB signature nevertheless maintained high accuracy for distinguishing ATB from NTB. In evaluation cohort E2, NTB samples were obtained from patients with bacterial pneumonia (BPN), fungal infections, or lung cancer (CA). Sensitivity and specificity for ATB diagnosis reached 88%, and although the AUC and sensitivity were comparable to those observed in ATB-vs-LTBI comparisons, specificity declined markedly from 96.3% to 87.7%. Among the non-TB diseases, lung cancer exhibited the greatest cross-reactivity, yielding false-positive ATB predictions in 15% of CA samples. This association may reflect the known comorbidity of the two diseases: a nationwide study documented simultaneous detection of lung cancer and active tuberculosis in Chinese patients(44) . Indeed, several individuals in our cohort were likewise IGRA-positive. In contrast, BPN cases were reliably excluded, resulting in a 100% negative predictive value.

Whole blood and PBMC samples are complex mixtures containing diverse immune cell subsets and numerous non-cellular components, each contributing to the overall transcriptomic signal. Consequently, the specific transcriptional signature of *M. tuberculosis* infection may be masked by unrelated background signals when DEGs are identified from bulk samples. Focusing on key immune cells could reduce this noise and detect more functionally related genes, thereby providing an opportunity to identify more sensitive signatures. Our results show that most DEGs in neutrophils are associated with immune response pathways, with over 180 DEGs related to annotated GO terms

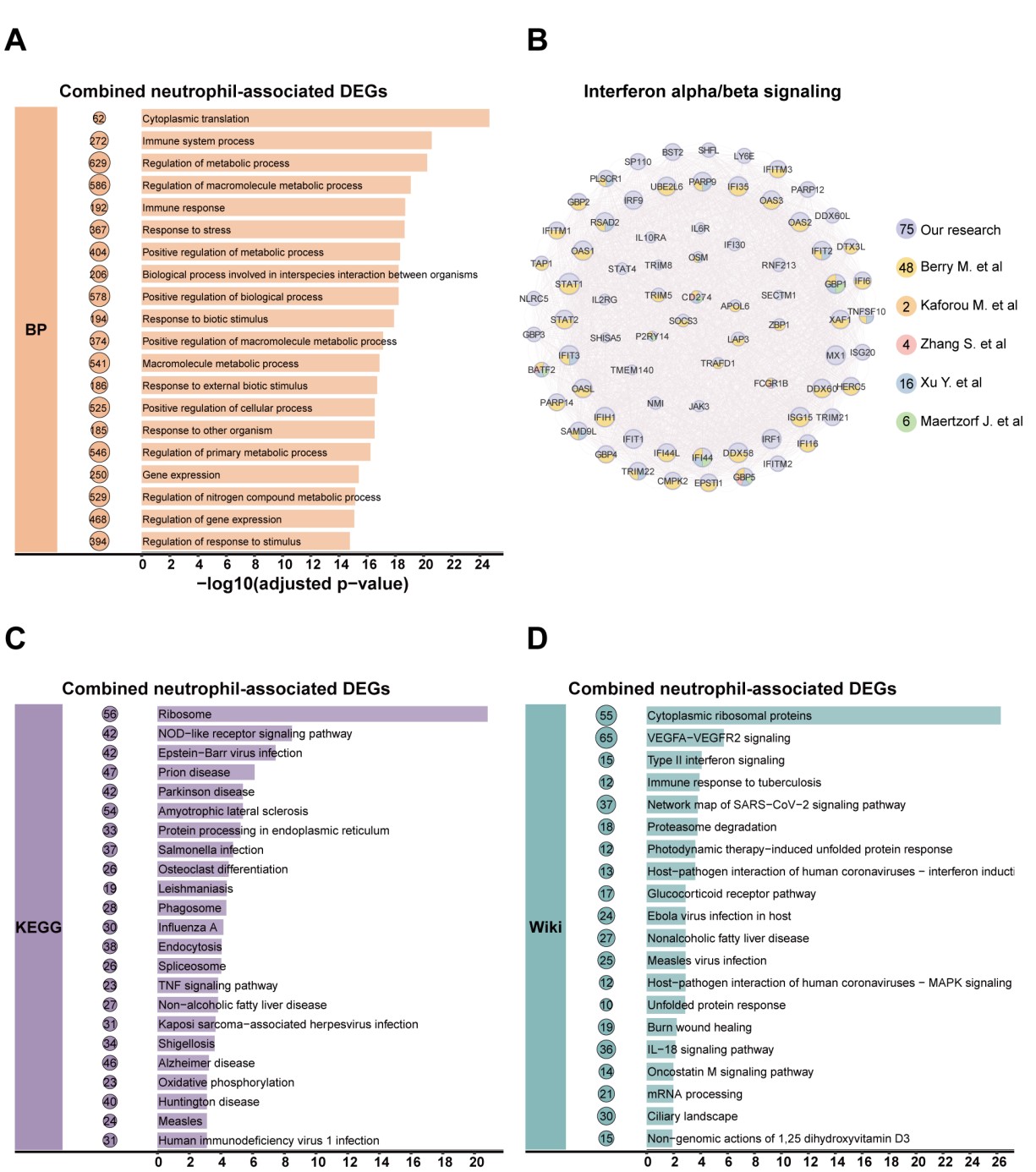

**FIG 6** Go enrichment analysis of combined DEGs in CD64[+] and CD15[+] neutrophil subsets. (A, C, and D) Functional annotation of the gene sets was performed using GO biological process, KEGG and Wiki pathway databases, respectively. (B) The interferon (IFN) signaling pathway cluster is shown; the numbers of IFN associated genes reported in previous studies are indicated by different colors.

(Fig. 6). Moreover, we identified 75 DEGs associated with the interferon-α/β pathway, an important pathway in TB pathogenesis. This number exceeds that reported for whole blood and PBMC samples in previous studies (10, 45–48). For example, 59 of the 393 DEGs reported by Berry et al. mapped to the IFN pathway, with 48 of these overlapping with our 75-gene cluster. In other studies, fewer than 20 IFN pathway DEGs were shared with our data set (Fig. 6B). Additionally, 25 DEGs were not reported in these studies, including those not reported in the IFN signal pathway, such as BST2 and LY6E.

Furthermore, the magnitude of differential gene expression was attenuated when qPCR was performed on whole-blood RNA compared with purified neutrophils. For the four-gene neutrophil-TB signature, only GBP5 remained significantly different between ATB and both LTBI and HC groups. In contrast, the other three genes lost their significance (Fig. S6). Consistent with this observation, the absolute expression level of GBP5 in whole blood was approximately one-quarter of that measured in neutrophils.

Consequently, the neu-TB signature reflects a more sensitive host response to ATB and holds promise for wider diagnostic application. Given that magnetic-activated cell sorting (MACS) is a well-established technology and qPCR is routinely available in both district and central hospitals, the signature is readily adaptable to automated platforms and holds immediate translational potential. A limitation of our study is that the signature's ability to discriminate ATB from other diseases remains to be validated in large, multicohort populations. Notably, non-tuberculous mycobacterial (NTM) infections, which can elevate the false-positive rate among controls, were not represented in the present cohort. In addition, the low yield of CD64[+] cells introduced stochastic noise that may compromise transcriptomic accuracy; therefore, systematic validation of these confounders in multiple, independent cohorts is a primary objective of our future work.

## ACKNOWLEDGMENTS

We would like to thank the patients and other participants for their cooperation in our study. We also thank the healthcare workers involved in the patient care and sample collection. Flow cytometry experiments were conducted at the Core Facilities and Service Centers, NIPB, CAMS&PUMC. We thank Dr. Li Li for expert assistance with experiments and analysis.

This study was supported by the CAMS Initiative for Innovative Medicine (2021-I2M-1-037, 2023-I2M-2-001).

J.H., S.L., and L.L. designed and conducted the experiments and collected data. J.H. wrote the manuscript. X.G., Q.Y., Q.C., and H.X. analyzed and interpreted the data. L.G. and X.Z. analyzed the results and reviewed the manuscript. Q.J. acquired the funding and supervised the study throughout the project. All authors have confirmed the authenticity of the raw data. All authors have read and approved the final version of the manuscript.

## AUTHOR AFFILIATIONS

[1]NHC Key Laboratory of Systems Biology of Pathogens, National Institute of Pathogen Biology, Chinese Academy of Medical Sciences & Peking Union Medical College, Beijing, People's Republic of China

[2]Center for Tuberculosis Research, Chinese Academy of Medical Sciences & Peking Union Medical College, Beijing, People's Republic of China

[3]Guangdong Key Lab for Diagnosis & Treatment of Emerging Infectious Diseases, Shenzhen Third People's Hospital, Shenzhen, People's Republic of China

[4]State Key Laboratory of Respiratory Health and Multimorbidity, Chinese Academy of Medical Sciences & Peking Union Medical College, Beijing, People's Republic of China

## AUTHOR ORCIDs

Lei Gao http://orcid.org/0000-0002-9957-4527
Xiaobing Zhang http://orcid.org/0009-0004-9037-2701
Qi Jin http://orcid.org/0000-0002-3586-6344

## FUNDING

| Funder | Grant(s) | Author(s) |
|---|---|---|
| CAMS Innovation Fund for Medical Sciences | 2023-I2M-2-001, 2021-I2M-1-037 | Qi Jin |

## AUTHOR CONTRIBUTIONS

Jie Hu, Data curation, Formal analysis, Methodology, Writing – original draft | Song Liu, Data curation, Methodology, Validation | Liguo Liu, Data curation, Methodology, Validation | Xingzhu Geng, Data curation, Formal analysis, Methodology, Software | Qianting Yang, Formal analysis, Methodology, Resources | Qi Chen, Data curation, Methodology, Software | Henan Xin, Formal analysis, Investigation, Validation | Lei Gao, Conceptualization, Investigation, Validation, Writing – review and editing | Xiaobing Zhang, Formal analysis, Methodology, Project administration, Validation, Writing – review and editing | Qi Jin, Conceptualization, Funding acquisition, Resources, Supervision

## DATA AVAILABILITY

The RNA sequencing data in this study can be found in the National Genomics Data Center (NGDC) under the accession number PRJCA010442.

## ETHICS APPROVAL

Studies involving human participants were reviewed and approved by The Ethics Committees of the Institute of Pathogen Biology, Chinese Academy of Medical Sciences. The patients provided written informed consent to participate in the study.

## ADDITIONAL FILES

The following material is available online.

### Supplemental Material

**Supplemental material (Spectrum01915-25-s0001.docx).** Fig. S1 to S6; Tables S1 and S2.

### Open Peer Review

**PEER REVIEW HISTORY (review-history.pdf).** An accounting of the reviewer comments and feedback.

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
