## [Reviewer comments · Microbiology Spectrum]

Microbiology Spectrum

Neutrophil-Specific Transcriptomic Profiling reveals a Novel Signature for Active Tuberculosis Diagnosis

Jie Hu, Song Liu, Liguo Liu, Xingzhu Geng, Qianting Yang, Qi Chen, Henan Xin, Lei Gao, Xiaobing Zhang, and Qi Jin

Corresponding Author(s): Xiaobing Zhang, National Institute of Pathogen Biology, Chinese Academy of Medical Sciences & Peking Union Medical College

Review Timeline:

Submission Date:	July 16, 2025
Editorial Decision:	October 17, 2025
Revision Received:	December 4, 2025
Editorial Decision:	December 24, 2025
Revision Received:	January 26, 2026
Accepted:	January 28, 2026

Editor: Sladjana Prisc

Reviewer(s): Disclosure of reviewer identity is with reference to reviewer comments included in decision letter(s). The following individuals involved in review of your submission have agreed to reveal their identity: A Christian Whelen (Reviewer #1)

Transaction Report:

DOI: <https://doi.org/10.1128/spectrum.01915-25>

Re: Spectrum01915-25 (Neutrophil-Specific Transcriptomic Profiling reveals a Novel Signature for Active Tuberculosis Diagnosis)

Dear Dr. Xiaobing Zhang:

Thank you for the privilege of reviewing your work. Below you will find my comments, instructions from the Spectrum editorial office, and the reviewer comments.

Revision Guidelines

Sincerely,
Sladjana Priscic
Editor
Microbiology Spectrum

Reviewer #1 (Comments for the Author):

Please see attachment for review

Reviewer #2 (Comments for the Author):

The manuscript by Hu et al describes work where they compared the transcriptional signatures of CD64+ and CD15+ neutrophils in patients with active TB, people with latent TB, and people with non-TB disease to define diagnostic transcriptional signatures for TB. The investigators flow sorted CD64+ cells from 2 mL of whole blood and used magnetic bead isolation to enrich a population of CD15+ neutrophils from 1 mL of blood. The authors then go through an analysis of the bulk transcriptional data from these subsets and show that there is a 4-gene signature that differentiates active and latent TB from each other and from healthy controls. The data are interesting on several levels both clinically and from a basic perspective on neutrophil biology in TB patients and a neutrophil-associated transcriptional signature would be a useful diagnostic tool at some point if validated. Overall, the manuscript is easy to read and a reasonable amount of information is provided on the methods used by the authors. Some questions and important issues need to be clarified, however.

- A minor clarification: the study design includes a substantial number of participants
- where the same assays on CD15+ and CD64+ cells done on each participant?
- The authors appear to use two different methods for RNA library prep. Is this impression accurate? If it isn't this accurate, this needs to be clarified in the M&M. If these methods are different, are there data suggesting they are complementary and equivalent?
- It is unclear how many cells, on average, were isolated from each population for sequencing. The Discussion suggests that a minimum of 150 cells per subset were analyzed (line 376), which is a very small number of cells for analysis. Some additional information on this factor should be included in the manuscript.
- The supplemental data for the CD64 separation suggests that an intermediate population was selected for RNA isolation. It doesn't appear that the sorted population is going to be clear of monocytes, which may also express CD64 at some level, or was a pure population of neutrophils. Can the authors comment on how clean their sorted population actually was?
- It isn't clear how clean the populations of CD15+ populations were after magnetic bead isolation. Some data demonstrating the purity of this population should be included.
- Do cells that express CD64 also express CD15? This seems likely, and presents an analytic problem for this study. Which population is the major population and if there's overlap, are the CD15+ cells a subset of the CD64+ population or vice versa? The authors should clarify this and show supporting data.
- I don't have any substantive comments on the analysis, which are interesting and, to this reviewer's level of expertise, appear to be properly done.
- Minor edits and typos:
 - Line 68: the authors cite the 2023 WHO report - an updated version of this report is available and should be cited instead
 - Line 80: M. tuberculosis should be abbreviated as M. tb as they indicate in line 70. Same issue for Line 440.
 - Table 1: Gender has a typo (Ginder)
 - Line 270: which genes are used as negative controls (specifically state them here)
 - Line 311 space missing between 0.885 and (Cohort E1)
 - Figure 6A is not referenced in the text

The manuscript “Neutrophil-Specific Transcriptomic Profiling reveals a Novel Signature for Active Tuberculosis Diagnosis” by Jie Hu, Song Liu, Liguang Liu, Xingzhu Geng, Qianting Yang, Qi Chen, Henan Xin, Lei Gao, Xiaobing Zhang, Qi Jin describes the use of a neutrophil-specific transcriptomic profile consisting of 4 genes to differentiate active tuberculosis, latent tuberculosis, and healthy controls. The authors isolated neutrophils in two different cohorts (CD64 and CD 15), then conducted transcriptomic profiling to identify candidate genes. Quantitative PCR was performed on RNA sequences to select and validate the 4 targets in CD15+ cells. The performance was then evaluated in 2 cohorts representing 5 clinical sites in 4 locations. The study is well-designed and executed. The data and analysis supports their conclusions with the possible exception of an overstatement. While I agree the strategy has diagnostic potential, the suggestion that this may meet triage or point of care criteria (stated in the “Importance” category of the manuscript submission) making it available for use by non-laboratorians in a patient care setting (e.g., Urgent Care or Emergency Department) is difficult to envision based on the described methods (neutrophil isolation, qPCR in an open system, data interpretation). Unless the authors better describe how this could be operationalized outside of a laboratory, recommend muting this claim until future advances make it more realistic.

In addition to this observation, here are a few additional points to address:

Major:

Line 79: This percentage is highly variable and heavily dependent on a multitude of factors such as prevalence, access to healthcare, availability of diagnostics like PCR, HIV status, living conditions, etc. This should be presented as a range with adequate references (Ref 5 appears to be a specific subpopulation and shouldn't be used to generalize the global percentage).

Line 330: A false positive rate of 15.2% (5 of 33) in NTB lung cancer patients is quite high. Another limitation is the lack of patients who had non-tuberculous mycobacterial (NTM) pulmonary disease, especially those that are AFB smear positive which could significantly impact “positive prediction rate ... for smear-positive isolates, reaching 100% in cohorts V and E2” (line 329-330; 399-400). Given the small number of NTB specimens in the study, lung cancer false positives, and the high prevalence of NTM infection in some populations, many of whom are AFB smear positive, recommend the authors address these limitations in more detail.

Line 339: The term “bacteriologically negative” is ambiguous. Need to explain/differentiate ATB in which the smear, PCR, and culture are negative versus LTBI in which smear, PCR, and culture are normally negative (not typically considered false negative) and the appropriate IGRA and/or skin test is often positive.

Minor:

Line 33: active and latent shouldn't be capitalized

Line 70, 72, 100: *M.tb* is not proper – *M. tuberculosis* is appropriate shortened taxonomy.

Line 80, 414: *M. tuberculosis* should be italicized.

Line 84: Point of care diagnostic do exist such as the Cepheid GeneXpert MTB-RIF.

Line 160, 166: What type of PCR instrument(s) were used?

Line 201: Fig 1. In the boxes, recommend putting categories of patients (ATB, LTBI, HC, NTB) in the same order.

Line 260: areindicated (missing space).

Line 329-330: Shouldn't "NTB isolates" be "NTB patients".

Line 385: *Salmonella* should be italicized.

Line 428: Signaling typo

The manuscript “Neutrophil Specific Transcriptomic Profiling reveals a Novel Signature for Active Tuberculosis Diagnosis” by Jie Hu, Song Liu, Liguu Liu, Xingzhu Geng, Qianting Yang, Qi Chen, Henan Xin, Lei Gao, Xiaobing Zhang, Qi Jin describes the use of a neutrophil specific transcriptomic profile consisting of 4 genes to differentiate active tuberculosis, latent tuberculosis, and healthy controls. The authors isolated neutrophils in two different cohorts (CD64 and CD 15), then conducted transcriptomic profiling to identify candidate genes. Quantitative PCR was performed on RNA sequences to select and validate the 4 targets in CD15+ cells. The performance was then evaluated in 2 cohorts representing 5 clinical sites in 4 locations. The study is well designed and executed. The data and analysis supports their conclusions with the possible exception of an overstatement. While I agree the strategy has diagnostic potential, the suggestion that this may meet triage or point of care criteria (stated in the “Importance” category of the manuscript submission) making it available for use by non laboratorians in a patient care setting (e.g., Urgent Care or Emergency Department) is difficult to envision based on the described methods (neutrophil isolation, qPCR in an open system, data interpretation). Unless the authors better describe how this could be operationalized outside of a laboratory, recommend muting this claim until future advances make it more realistic.

Response: I fully agree with the suggestion and have added the corresponding statement to the Discussion.

Line 437-441: “Given that magnetic-activated cell sorting (MACS) is a well-established technology and qPCR is routinely available in both district and central hospitals, the signature is readily adaptable to automated platforms and holds immediate translational potential. This assay has the potential to meet triage or point-of-care criteria,

enabling future deployment by non-laboratory personnel in clinical settings.”

In addition to this observation, here are a few additional points to address:

Major:

Line 79: This percentage is highly variable and heavily dependent on a multitude of factors such as prevalence, access to healthcare, availability of diagnostics like PCR, HIV status, living conditions, etc. This should be presented as a range with adequate references (Ref 5 appears to be a specific subpopulation and shouldn't be used to generalize the global percentage).

Response: I fully agree with the suggestion and have revised the statement accordingly. “...approximately 50% of clinical cases fail to yield conclusive bacteriological evidence (5)” as in Line 76-79, meanwhile other two references were added in this sentence.(previous version)

Line 80-84: “..., 40-70% of clinical cases remain bacteriologically unconfirmed (negative smear, culture and molecular assays). This wide variability reflects diverse determinants—background pathogen prevalence, health-care access, availability of molecular diagnostics (e.g., PCR), HIV co-infection, living conditions, and other contextual factors (5-7).”

Line 330: A false positive rate of 15.2% (5 of 33) in NTB lung cancer patients is quite high. Another limitation is the lack of patients who had non-tuberculous mycobacterial (NTM)

pulmonary disease, especially those that are AFB smear positive which could significantly impact “positive prediction rate ... for smear-positive isolates, reaching 100% in cohorts V and E2” (line 329-330; 399-400). Given the small number of NTB specimens in the study, lung cancer false positives, and the high prevalence of NTM infection in some populations, many of whom are AFB smear positive, recommend the authors address these limitations in more detail.

Response: The relatively high false-positive rate (15.2 %) in the NTB group may reflect co-existing *M. tuberculosis* infection; several of these patients were IGRA-positive. Because nontuberculous mycobacteria (NTM) were not specifically examined, I plan to include them in future validation studies. A statement describing this limitation has been added to the Discussion.

Lin 411-414: “This association may reflect the known comorbidity of the two diseases: a nationwide study documented simultaneous detection of lung cancer and active tuberculosis in Chinese patients (43). Indeed, several individuals in our cohort were likewise IGRA-positive.”

Line 443-445: “Notably, non-tuberculous mycobacterial (NTM) infections, which can elevate the false-positive rate among controls, were not represented in the present cohort. Systematic validation against such confounding diseases across multiple, independent populations is a primary.

Line 339: The term “bacteriologically negative” is ambiguous. Need to explain/differentiate ATB in which the smear, PCR, and culture are negative versus LTBI in which smear, PCR, and culture are normally negative (not typically considered false negative) and the appropriate IGRA and/or skin test is often positive.

Response: The previous sentence (“Current sputum-based diagnostic approaches face inherent limitations due to the inability of existing methods to reliably diagnose

bacteriologically negative TB cases or individuals from whom it is difficult to acquire sputum specimens.”) was replaced by this:

Line 323-325: “Sputum-based diagnostics cannot reliably detect TB in patients who are smear- or culture- negative yet IGRA- or TST- positive, nor in those unable to expectorate.”

Minor:

Line 33: active and latent shouldn't be capitalized

Response:

Line 33: the word of “Active” and “Latent” were changed as “active” and “latent”

Line 70, 72, 100: *M.tb* is not proper – *M. tuberculosis* is appropriate shortened taxonomy. Line 80, 414: *M. tuberculosis* should be italicized.

Response:

Line 72: “*Mycobacterium tuberculosis*”;

Line 74, 85, 105, 418: the appropriate form of “*M. tuberculosis*” was placed in these sentences.

Line 84: Point of care diagnostic do exist such as the Cepheid GeneXpert MTB-RIF.

Response:

Line 89-90: the sentence “Moreover, current diagnostic paradigms lack point-of-care testing solutions that combine operational simplicity and rapid turnaround times.” was deleted from the introduction.

Line 160, 166: What type of PCR instrument(s) were used?

Response: the information about PCR instrument was added in M&M.

Line 185-187: “qPCR validation of host target-gene expression was carried out with TaqMan® Fast Advanced Master Mix (Thermo Fisher Scientific) on the Applied Biosystems™ QuantStudio™ 7 Pro Real-Time PCR System (Thermo Fisher Scientific).

Line 201: Fig 1. In the boxes, recommend putting categories of patients (ATB, LTBI, HC, NTB) in the same order.

Response:

Fig. 1: The patient categories have been reordered to match the sequence used throughout the manuscript.

Line 260: areindicated (missing space).

Response:

Line 488: Fig 3: areindicated” is replaced by “are-indicated” in Legend

Line 329-330: Shouldn’t “NTB isolates” be “NTB patients”.

Response:

Line 315: “...NTB isolates...” is replaced by “NTB patients”

Line 385: *Salmonella* should be italicized.

Response:

Line 385: “Salmonella” is replaced by “*Salmonella*”

Line 428: Signaling typo

Response:

Line 510: Fig 6 Legend: the word “signalling” is replaced by “signaling”.

Response to Reviewers:

The manuscript by Hu et al describes work where they compared the transcriptional signatures of CD64⁺ and CD15⁺ neutrophils in patients with active TB, people with latent TB, and people with non-TB disease to define diagnostic transcriptional signatures for TB. The investigators flow sorted CD64⁺ cells from 2 mL of whole blood and used magnetic bead isolation to enrich a population of CD15⁺ neutrophils from 1 mL of blood. The authors then go through an analysis of the bulk transcriptional data from these subsets and show that there is a 4-gene signature that differentiates active and latent TB from each other and from healthy controls. The data are interesting on several levels both clinically and from a basic perspective on neutrophil biology in TB patients and a neutrophil-associated transcriptional signature would be a useful diagnostic tool at some point if validated. Overall, the manuscript is easy to read and a reasonable amount of information is provided on the methods used by the authors. Some questions and important issues need to be clarified, however.

- A minor clarification: the study design includes a substantial number of participants

Response: In this study, a cross-section analysis was performed and there were 646 participants included in 5 cohorts.

Line 217: "...646 clinical isolates..." was replaced by "...clinical isolates from 646 participants..."

- where the same assays on CD15⁺ and CD64⁺ cells done on each participant?

Response: In our study, CD15⁺ and CD64⁺ neutrophils were isolated from distinct participant cohorts; each individual contributed cells for only one subset, and no participant underwent both assays. To clarify this design, two sentences have been added to the Materials and Methods.

Line 136-137: "In this study, equal numbers of CD64⁺ cells were isolated from each of 114 participants (discovery cohort D1: 52 ATB, 19 LTBI, 43 HC) and subjected to RNA sequencing."

Line 152-154: "Differential gene expression was analyzed separately for two RNA-seq datasets: the newly generated CD64⁺ subset profiles and our previously published CD15⁺ subset data (Discovery cohort D2: 15 ATB, 23 LTBI, 21 HC) (13)."

- The authors appear to use two different methods for RNA library prep. Is this impression accurate? If it isn't this accurate, this needs to be clarified in the M&M. If these methods are different, are there data suggesting they are complementary and equivalent?

Response: Two library-preparation kits were employed: the SMARTer Stranded Total

RNA-Seq Kit v3-Pico Input Mammalian (CD64⁺ subset) and the SMARTer Stranded Total RNA-Seq Kit v2 (CD15⁺ subset). Libraries were sequenced and analyzed with identical pipelines, and the resulting DEGs were combined for downstream pathway analysis. This strategy ensures the two datasets are equivalent and complementary; a paragraph describing these details has been added to the Materials and Methods.

Line 152-157: “Differential gene expression was analyzed separately for two RNA-seq datasets: the newly generated CD64⁺ subset profiles and our previously published CD15⁺ subset data (Discovery cohort D2: 15 ATB, 23 LTBI, 21 HC) (13). RNA-seq libraries of CD15⁺ subset were generated with the SMARTer® Stranded Total RNA-Seq Kit v2 (Takara) from 0.5 ml of participant peripheral blood. Sequencing was performed on the NovaSeq 6000 platform.”

- It is unclear how many cells, on average, were isolated from each population for sequencing. The Discussion suggests that a minimum of 150 cells per subset were analyzed (line 376), which is a very small number of cells for analysis. Some additional information on this factor should be included in the manuscript.

Response: To improve clarity, additional experimental details have been incorporated into both the Materials and Methods and the Discussion sections:

Line 135: The previous title “CD64⁺ neutrophil isolation and RNA sequencing” was replaced by “Transcriptomic Profiling of Equal Numbers of CD64⁺ Cells from Clinical Samples”.

Line 142-143: “A total of 150 CD45⁺CD3⁻CD64^{dim} cells per sample were sorted into EP buffer...”

Line 147-149: “Using a small number of collected cells minimized cellular degradation, while the all-in-one SMARTer kit protocol enhanced experimental precision across all isolates.”

Line 371-373: “Neutrophils are notoriously labile in the peripheral circulation; enriching only a small number of cells limited degradation and ensured experimental precision.”

- The supplemental data for the CD64 separation suggests that an intermediate population was selected for RNA isolation. It doesn't appear that the sorted population is going to be clear of monocytes, which may also express CD64 at some level, or was a pure population of neutrophils. Can the authors comment on how clean their sorted population actually was?

Response:

To identify a transcriptomic signature for ATB diagnosis, we isolated total CD64⁺ cells as the RNA-seq target population. Liu QQ et al. previously demonstrated that CD64⁺ neutrophils expand selectively in ATB, whereas the CD64⁺ monocyte fraction remains small and stable. We therefore assumed that the differentially expressed genes (DEGs) detected in our bulk RNA-seq data would largely reflect changes in CD64⁺ neutrophils. Flow-cytometry results confirmed that

the percentage of CD64⁺ cells increases during M. tb infection (Supplementary Fig. S2). A corresponding statement has now been added to the Discussion (lines 349-351).

We agree that isolating CD64⁺ neutrophils and CD64⁺ monocytes separately for RNA-seq would provide higher resolution and we plan to pursue this approach in future work

Supplementary:

Figure S2. Statistical analysis of CD64⁺ subset expansion across participant groups in Discovery Cohort 1. The numbers of CD64⁺ cells were significantly higher in both ATB and LTBI groups than in HC ($p < 0.0001$), indicating that expansion of the CD64⁺ subset is closely associated with M. tuberculosis infection and disease status.

Line 361-363: “Flow-cytometric quantification in cohort D1 showed that the CD64⁺ compartment was significantly expanded in both ATB and LTBI isolates compared with HC samples (Supplementary Figure S2).”

- It isn't clear how clean the populations of CD15⁺ populations were after magnetic bead isolation. Some data demonstrating the purity of this population should be included.

Response: According to the manufacturer (Thermo Fisher Scientific), the CD15⁺ granulocyte isolation procedure routinely yields >95 % pure cells. This information has been added to the M&M.

Line 177-178: “This protocol consistently recovers >95 % of CD15⁺ granulocytes from whole-blood samples (Thermo Fisher Scientific data).”

- Do cells that express CD64 also express CD15? This seems likely, and presents an analytic

problem for this study. Which population is the major population and if there's overlap, are the CD15+ cells a subset of the CD64+ population or vice versa? The authors should clarify this and show supporting data.

Response: To determine whether the CD64+ subset also expresses CD15, additional flow-cytometry analysis was performed. The majority of target cells co-expressed CD64 and CD15, confirming their neutrophil identity; this finding is now described in the Discussion and supported by data shown in Supplementary Fig. S4.

Line 363-366: “Further analysis revealed robust co-expression of CD15 and CD64 on neutrophils in ATB isolates. Dual staining of peripheral-blood leukocytes demonstrated that >90 % of CD64+ cells co-expressed CD15, and conversely the majority of CD15+ cells fell within the CD64+ gate (Supplementary Figure S3).”

Line 375-378: “KEGG analysis of the DEGs revealed a shared transcriptional signature between the two subsets: 60 % (ATB vs HC) and 69 % (ATB vs LTBI) of CD64+ DEGs were concordantly identified in CD15+ cells (Supplementary Figure S4), indicating that both subset-specific analyses converge on the same analytical target.”

- I don't have any substantive comments on the analysis, which are interesting and, to this reviewer's level of expertise, appear to be properly done.

- Minor edits and typos:

- Line 68: the authors cite the 2023 WHO report- an updated version of this report is available and should be cited instead

Response:

Line 70: the new version of 2024 WHO report was updated.

- Line 80: M. tuberculosis should be abbreviated as M. tb as they indicate in line 70.

Same issue for Line 440.

Response:

Line 72, 418: The phrases “M. tuberculosis” in these two sentences were replaced by “*M. tuberculosis*”.

- Table 1: Gender has a typo (Ginder)

Response:

Table 1: The typo (Ginder) has been replaced by Gender.

- Line 270: which genes are used as negative controls (specifically state them here)

Response:

Line 271: Two genes (NFE2 and THEMIS2) were added in this sentence, which were used as negative controls.

Re: Spectrum01915-25R1 (Neutrophil-Specific Transcriptomic Profiling reveals a Novel Signature for Active Tuberculosis Diagnosis)

Dear Dr. Xiaobing Zhang:

Thank you for the privilege of reviewing your work. Below you will find my comments, instructions from the Spectrum editorial office, and the reviewer comments.

Revision Guidelines

Sincerely,
Sladjana Priscic
Editor
Microbiology Spectrum

Reviewer #1 (Comments for the Author):

Please see attached review

Reviewer #2 (Comments for the Author):

The authors' revisions have improved the manuscript but several items should be addressed to improve it further.

This includes:

- The authors rely on the manufacturer's indication for purity of their populations by magnetic bead separation. Although this approach may yield that level of purity, there can be substantial variability and this should be noted as a limitation if the authors are not showing data for the range of purities they recover.
- Likewise, the small number of cells used for some of their transcriptomic assays is a limitation. The authors indicate this was done because of the labile nature of neutrophils but neutrophils can be difficult to obtain RNA from and data from a small number of cells is susceptible to increased error when even a fraction of this population is not pure. Considering these issues, this should be described as a limitation in the Discussion.
- Figure S3 needs to be labeled so the reader knows what antigen is being assayed on each panel and which populations are being analyzed.

The manuscript “Neutrophil-Specific Transcriptomic Profiling reveals a Novel Signature for Active Tuberculosis Diagnosis” by Jie Hu, et. al. claims that this may meet triage or point of care criteria making it available for use by non-laboratorians in a patient care setting (e.g., Urgent Care or Emergency Department). This is difficult to envision based on the described methods (neutrophil isolation, qPCR in an open system, data interpretation). Unless the authors better describe how this could be operationalized outside of a laboratory, recommend muting this claim until future advances make it more realistic.

Response: I fully agree with the suggestion and have added the corresponding statement to the Discussion.

Line 437-441: “Given that magnetic-activated cell sorting (MACS) is a well-established technology and qPCR is routinely available in both district and central hospitals, the signature is readily adaptable to automated platforms and holds immediate translational potential. This assay has the potential to meet triage or point-of-care criteria, enabling future deployment by non-laboratory personnel in clinical settings.”

Reviewer assessment: **Further revision required.** Nothing in the paper supports the use of this assay as triage or point-of-care (POC) by non-lab personnel. The last sentence should be deleted as well as the POC reference in line 67.

In addition to this observation, here are a few additional points to address:

Major:

Line 79: This percentage is highly variable and heavily dependent on a multitude of factors such as prevalence, access to healthcare, availability of diagnostics like PCR, HIV status, living conditions, etc. This should be presented as a range with adequate references (Ref 5 appears to be a specific subpopulation and shouldn't be used to generalize the global percentage).

Response: I fully agree with the suggestion and have revised the statement accordingly. “...approximately 50% of clinical cases fail to yield conclusive bacteriological evidence (5)” as in Line 76-79, meanwhile other two references were added in this sentence.(previous version)

Line 80-84: “..., 40-70% of clinical cases remain bacteriologically unconfirmed (negative smear, culture and molecular assays). This wide variability reflects diverse determinants—background pathogen prevalence, health-care access, availability of molecular diagnostics (e.g., PCR), HIV co-infection, living conditions, and other contextual factors (5-7).”

Reviewer assessment: **Further revision required**. The 2 additional references are a second Chinese study and one in Uganda. That is not global representation and makes the lower end of the range (40%) way too high. Developed countries do not miss 40% of cases because of false negating testing, and even some resource constrained countries now have access to GeneXpert, which has significantly improved diagnostics over unconcentrated AFB Stain. The authors can either define this limited population (not recommended) or reference global data such as that available from the World Health Organization to determine more accurately the lower limit of the range.

Line 330: A false positive rate of 15.2% (5 of 33) in NTB lung cancer patients is quite high. Another limitation is the lack of patients who had non-tuberculous mycobacterial (NTM) pulmonary disease, especially those that are AFB smear positive which could significantly impact “positive prediction rate ... for smear-positive isolates, reaching 100% in cohorts V and E2” (line 329-330; 399-400). Given the small number of NTB specimens in the study, lung cancer false positives, and the high prevalence of NTM infection in some populations, many of whom are AFB smear positive, recommend the authors address these limitations in more detail.

Reviewer assessment: **Acceptable** response.

Line 339: The term “bacteriologically negative” is ambiguous. Need to explain/differentiate ATB in which the smear, PCR, and culture are negative versus LTBI in which smear, PCR, and culture are normally negative (not typically considered false negative) and the appropriate IGRA and/or skin test is often positive.

Reviewer assessment: Minor **wording change** is necessary. Change “**Sputum-based diagnostics cannot reliably detect TB...**” to “Sputum-based diagnostics lack sensitivity to detect TB...”. Authors should also spell out “smear-negative” rather than smear-.

Minor:

Reviewer assessment: All responses are **acceptable**.

Response to Reviewer's comments:

ASM Microbiology Spectrum 01915-25R1

The manuscript “Neutrophil-Specific Transcriptomic Profiling reveals a Novel Signature for Active Tuberculosis Diagnosis” by Jie Hu, et. al. claims that this may meet triage or point of care criteria making it available for use by non-laboratorians in a patient care setting (e.g., Urgent Care or Emergency Department). This is difficult to envision based on the described methods (neutrophil isolation, qPCR in an open system, data interpretation). Unless the authors better describe how this could be operationalized outside of a laboratory, recommend muting this claim until future advances make it more realistic.

Response: I fully agree with the suggestion and have added the corresponding statement to the Discussion.

Line 437-441: “Given that magnetic-activated cell sorting (MACS) is a well-established technology and qPCR is routinely available in both district and central hospitals, the signature is readily adaptable to automated platforms and holds immediate translational potential. This assay has the potential to meet triage or point-of-care criteria, enabling future deployment by non-laboratory personnel in clinical settings.”

Reviewer assessment: **Further revision required.** Nothing in the paper supports the use of this assay as triage or point-of-care (POC) by non-lab personnel. The last sentence should be deleted as well as the POC reference in line 67.

Further revision: Thank you for this helpful suggestion.

We have removed the final sentence and also deleted the statement “This assay has the potential to meet triage or point-of-care criteria, enabling future deployment by non-laboratory personnel in clinical settings”

Line 63-64: The sentence “3). Application potential: The qPCR based assay aligns with WHO target product profiles for point of care TB diagnostics, requiring only 50 µL whole blood.” was deleted from the R2 version.

Line 420-422: the sentence “This assay has the potential to meet triage or point-of-care criteria, enabling future deployment by non-laboratory personnel in clinical settings.” was deleted from the R2 version.

In addition to this observation, here are a few additional points to address:

Major:

Line 79: This percentage is highly variable and heavily dependent on a multitude of factors such as prevalence, access to healthcare, availability of diagnostics like PCR,

HIV status, living conditions, etc. This should be presented as a range with adequate references (Ref 5 appears to be a specific subpopulation and shouldn't be used to generalize the global percentage).

Response: I fully agree with the suggestion and have revised the statement accordingly. "...approximately 50% of clinical cases fail to yield conclusive bacteriological evidence (5)" as in Line 76-79, meanwhile other two references were added in this sentence.(previous version)

Line 80-84: "..., 40-70% of clinical cases remain bacteriologically unconfirmed (negative smear, culture and molecular assays). This wide variability reflects diverse determinants—background pathogen prevalence, health-care access, availability of molecular diagnostics (e.g., PCR), HIV co-infection, living conditions, and other contextual factors (5-7)."

Reviewer assessment: **Further revision required.** The 2 additional references are a second Chinese study and one in Uganda. That is not global representation and makes the lower end of the range (40%) way too high. Developed countries do not miss 40% of cases because of false negating testing, and even some resource constrained countries now have access to GeneXpert, which has significantly improved diagnostics over unconcentrated AFB Stain. The authors can either define this limited population (not recommended) or reference global data such as that available from the World Health Organization to determine more accurately the lower limit of the range.

Further revision: We have updated the global epidemiology data with the 2025 WHO Global tuberculosis report. Furthermore, the previous references (5, 6) were replaced with three more recent ones, which as three other new references, which The revised sentences as:

Line 78-82: The previous sentence "...40-70% of clinical cases remain bacteriologically unconfirmed (negative smear, culture and molecular assays)." was revised as

"...only 64 % of the 6.9 million pulmonary TB cases reported worldwide in 2024 were bacteriologically confirmed; the remaining 19-45% were microbiologically unconfirmed (negative smear, culture and molecular assays), with the exact proportion varying by country or region."

Line 82-84 : New references (5–8) have been added to support this statement

Line 330: A false positive rate of 15.2% (5 of 33) in NTB lung cancer patients is quite high. Another limitation is the lack of patients who had non-tuberculous mycobacterial (NTM) pulmonary disease, especially those that are AFB smear positive which could significantly impact "positive prediction rate ... for smear-positive isolates, reaching 100% in cohorts V and E2" (line 329-330; 399-400). Given the small number of NTB specimens in the study, lung cancer false positives, and the

high prevalence of NTM infection in some populations, many of whom are AFB smear positive, recommend the authors address these limitations in more detail.

Reviewer assessment: **Acceptable** response.

Line 339: The term “bacteriologically negative” is ambiguous. Need to explain/differentiate ATB in which the smear, PCR, and culture are negative versus LTBI in which smear, PCR, and culture are normally negative (not typically considered false negative) and the appropriate IGRA and/or skin test is often positive.

Reviewer assessment: Minor **wording change** is necessary. Change “Sputum-based diagnostics cannot reliably detect TB...” to “Sputum-based diagnostics lack sensitivity to detect TB...”. Authors should also spell out “smear-negative” rather than smear-.

Further revision:

Line 311: the sentence “Sputum-based diagnostics cannot reliably detect TB...” was revised as “Sputum-based diagnostics lack sensitivity to detect TB...” in R2 version.

Minor:

Reviewer assessment: All responses are **acceptable**.

Response to Reviewers:

The authors' revisions have improved the manuscript but several items should be addressed to improve it further.

This includes:

- The authors rely on the manufacturer's indication for purity of their populations by magnetic bead separation. Although this approach may yield that level of purity, there can be substantial variability and this should be noted as a limitation if the authors are not showing data for the range of purities they recover.

Response: Thank you for this constructive suggestion.

In response, we have added a rigorous flow-cytometric purity assessment in Supplementary (new Fig. S2). Following immunomagnetic enrichment of CD15⁺ neutrophils with Dynabeads® CD15, whole blood (input), the post-selection fraction and the flow-through from a representative donor were stained in parallel with anti-CD45, anti-CD15 and anti-CD16 antibodies and analyzed on the same cytometer. 98.1% of the CD45⁺ events in the post-selection fraction were CD15⁺CD16⁺, confirming the high purity of the isolated neutrophils. The corresponding methodological details and results have been incorporated into the Materials and methods section and Fig. S2 legend.

---Line 170-172: the previous sentence “This protocol consistently recovers >95 % of CD15⁺ granulocytes from whole-blood samples (Thermo Fisher Scientific data).” was replaced by “This protocol consistently yields preparations in which > 90% of the enriched cells are CD15⁺ neutrophils from whole-blood samples (Supplementary Figure S2).”

- Likewise, the small number of cells used for some of their transcriptomic assays is a limitation. The authors indicate this was done because of the labile nature of neutrophils but neutrophils can be difficult to obtain RNA from and data from a small number of cells is susceptible to increased error when even a fraction of this population is not pure. Considering these issues, this should be described as a limitation in the Discussion.

Response: We appreciate the suggestion and have incorporated the revised description in both the Materials and methods and Discussion sections. We have now inserted the following

sentences at line 335 and of the revised manuscript:

---Line 138-141: “To minimize neutrophil RNA degradation, we adopted a single-tube “all-in-one” strategy: A total of 150 CD45+CD3-CD64dim cells per sample were directly sorted into RNase-Inhibitor-containing water and immediately processed for cDNA library construction.”

---Line 144-146: The sentence ‘Using a small number of collected cells minimized cellular degradation, while the all-in-one SMARTer kit protocol enhanced experimental precision across all isolates.’ was deleted.

---Line 425-428: “In addition, the low yield of CD64+ cells introduced stochastic noise that may compromise transcriptomic accuracy; ...”

- Figure S3 needs to be labeled so the reader knows what antigen is being assayed on each panel and which populations are being analyzed.

Response: We have revised Figure S3 and now label each panel to clarify the gating strategy in R2 version.

Re: Spectrum01915-25R2 (Neutrophil-Specific Transcriptomic Profiling reveals a Novel Signature for Active Tuberculosis Diagnosis)

Dear Dr. Xiaobing Zhang:

Please proofread the manuscript carefully (e.g., Line 141 - "construction" not "constriction", Line 311 - "smear-negative" not "smear-", as Reviewer 1 suggested).

Your manuscript has been accepted, and I am forwarding it to the ASM production staff for publication. Your paper will first be checked to make sure all elements meet the technical requirements. ASM staff will contact you if anything needs to be revised before copyediting and production can begin. Otherwise, you will be notified when your proofs are ready to be viewed.

Sincerely,
Sladjana Priscic
Editor
Microbiology Spectrum